# Evaluation of Endospore-Forming Bacteria for Suppression of Postharvest Decay of Apple Fruit

**DOI:** 10.3390/microorganisms11010081

**Published:** 2022-12-28

**Authors:** Anissa Poleatewich, Paul Backman, Haley Nolen

**Affiliations:** 1Department of Agriculture, Nutrition, and Food Systems, University of New Hampshire, Durham, NH 03824, USA; 2Department of Plant Pathology and Environmental Microbiology, Penn State University, University Park, PA 16802, USA; 3Department of Molecular, Cellular, and Biomedical Sciences, University of New Hampshire, Durham, NH 03824, USA

**Keywords:** *Bacillus*, bitter rot, blue mold, biological control, apple

## Abstract

The use of microbial biocontrol agents for control of postharvest disease has been the subject of intensive research over the past three decades resulting in commercialization of several biocontrol products. The objective of this research was to test endospore-forming bacteria collected from apple leaves for suppression of bitter rot and blue mold on apple. Bacteria were collected from abandoned, low-input, organic, and conventionally managed orchards in Pennsylvania and were screened for their ability to produce endospores, hydrolyze chitin, reduce pathogen growth in vitro, and suppress postharvest disease in vivo. Several isolates reduced bitter rot lesion size on ‘Rome Beauty’ from 40–89% compared to untreated controls. *Bacillus megaterium* isolates, A3-6 and Ae-1, resulted in the greatest suppression of bitter rot lesion size. One isolate, A3-2, suppressed blue mold lesion size. Scanning electron microscopy of inoculated apple wounds suggests parasitism as a mode of action explains the suppression of bitter rot lesion size by isolate A3-6. Of the top seventeen isolates exhibiting biocontrol potential, 70% were collected from abandoned or unmanaged locations. This research demonstrates abandoned apple orchards can be a source of new biocontrol agents for control of postharvest diseases of apple.

## 1. Introduction

Production of fresh fruits and vegetables presents a unique challenge as most produce is harvested over a relatively short period of time and then stored for weeks to months to avoid exceeding demand and to provide consumers with a product year-round. Postharvest decay by microorganisms represents a significant limitation in our ability to store fruit [1]. The economic losses incurred from storage diseases are considerable given the initial investment related to crop production and management of pathogens and pests in the field prior to harvest. To date, the use of synthetic fungicides (as pre- or postharvest treatments) remains an important strategy for managing postharvest decays [2,3]. Development of fungicide resistant strains has been documented for several postharvest pathogens [4,5]. Human health concerns, due to pesticide residues on surfaces directly consumed as food, have also limited the available options for control of postharvest pathogens [6,7]. Several countries have banned or significantly restricted postharvest applications of fungicides [3,8]. As a result, there is an ongoing effort to develop alternative control methods to reduce fungicide dependence, reduce environmental risks, and improve consumer confidence in food safety.

Considerable attention has focused on the use of biocontrol for management of postharvest diseases of fruits and vegetables, and there have been several review articles on the topic over the past 30 years [1,2,3,9,10,11,12,13]. The uniqueness of the postharvest environment offers an advantage as environmental variables are more stable compared to field conditions, and to an extent, parameters can be adjusted to favor survival of biocontrol agents (BCAs). The challenge is new products must not only be safe but effective and economical. Several microorganisms have been patented for postharvest biocontrol, many which are applied to harvested fruit as a dip or drench. Over the past two decades, postharvest biocontrol has evolved into a significant area of research. Pioneering work conducted by Wilson and Pusey [12,14] in the 1980s soon led to a wealth of research and development of commercial products. Although numerous products were pursued, most were met with limited success [2,9]. Currently, there are four commercial products available for postharvest use in the United States [15]: Bio-Save™ 10 and 11 containing two strains of *Pseudomonas syringae* (JET Harvest Solutions, Longwood, FL, USA), Lalfresh S containing *Clonostachys rosea* strain J1446 (*Gliocladium catenulatum*) (Lallemand Plant Care, Montreal, QC, Canada)., and the yeast-based product NEXY (Agrauxine by Lesaffre) containing *Candida oleophila* strain O. In their 2016 review, Droby and colleagues [9] highlighted several reasons contributing to poor market longevity of postharvest biocontrol products, including the need to consider the microhabitat in which BCAs are expected to reside when attempting to select and establish BCAs. Colonization of fruit surfaces and interaction with existing microflora are important considerations when screening BCAs [16]. Because of their popularity as biocontrol agents for soil-borne diseases [17], many researchers have attempted to employ soil/root-inhabiting *Bacillus* strains against foliar diseases without proper consideration of the necessary adaptations to survive in the phyllosphere or carposphere [18]. Therefore, selection of bacteria adapted to the target niche may increase the likelihood of long-term colonization and effective control of postharvest diseases. For example, BCAs for postharvest application should have the ability to survive at low temperatures typical of cold storage conditions [1]. Additionally, sampling from niches that are representative of the target application niche [19] may increase efficacy. For this reason, apple orchards may serve as a source for novel BCAs for postharvest biocontrol on apple. Furthermore, abandoned apple orchards—no longer exposed to chemical pesticides—may serve as a resource to identify novel BCAs or consortia of biocontrol microbes [20].

The objectives of this study were to (1) collect native bacteria from apple fruit and foliage from managed and abandoned apple trees, (2) screen these isolates for chitinase production, dual-culture antagonism, and suppression of fruit rots in vivo, and (3) further characterize the efficacy and modes of action of candidate isolates using scanning electron microscopy.

## 2. Materials and Methods

### 2.1. Isolation and Culture of Biocontol Bacteria

Healthy apple (*Malus domestica*) leaves and fruit were collected randomly from three orchards maintained under a standard fungicide, a ‘reduced risk’ fungicide program, no spray, or a certified organic program at the Penn State University Fruit Research and Extension Center (PSU FREC). Samples were also collected from feral trees in forested areas in Centre County, PA and from three abandoned orchards (two orchards unmanaged for one year and one orchard unmanaged for five years) in Adam’s County, PA in September 2006 and June 2007. Within the conventionally managed orchards at the PSU FREC, samples were collected from cv. Golden Delicious, Rome Beauty, and Red Delicious trees grown on Malling 26 (M.26) rootstock. Samples were also collected from cv. GoldRush and Enterprise grown on M.26 rootstock in an organic demonstration orchard. Cultivars in the abandoned orchards and feral areas were unknown. Samples were stored in a cooler with ice during transport to the laboratory. 

For isolation of bacteria, two leaves or two plugs of fruit tissue (collected with a 44.2 mm diameter cork borer) were placed in sterile 101 mm × 152 mm stomacher filter bags (Secure-T 80; Labplas, Sainte-Julie, QC, Canada) with 10 mL of 0.1 M potassium phosphate buffer and agitated at seven strokes per second for 30 s in a stomacher blender (Bagmixer 100 MiniMix; Intersciences St. Nom la Bretèche France). The leaf tissue was further triturated using a pestle. 50 µL of the suspension was dilution plated on each of two media, tryptic soy agar (TSA) amended with Benomyl and apple juice yeast extract agar (AJYE), using a spiral plater (Autoplate 4000; Spiral Biotech Inc., Norwood, MA, USA). To select for endospore-forming bacteria, three 1 mL aliquots of the agitated solution were pipetted into 1.7 mL microtubes and heat treated in a 75 °C water bath for 15 min. The heated solutions were plated on tryptic soy agar (TSA) and apple juice-yeast extract agar (AJYA). After incubation, bacterial colonies were sub-cultured onto yeast extract dextrose calcium carbonate agar (YDC) to obtain pure cultures.

### 2.2. Preliminary Screening for Antagonism and Disease Suppression

#### 2.2.1. Chitin Hydrolysis Assay

The bacteria isolated from apple leaves and fruit were first characterized for the ability to hydrolyze chitin using a conventional plate method. Colloidal chitin was prepared using a modified method of Kokalis-Burelle [21]. The percent chitin (*w*/*v*) of the suspension was determined using the oven dry weight of a 10 mL aliquot of the wet suspension. Aliquots of the settled chitin suspension were then blended for 30 s in a sterilized Waring laboratory blender (Model 7010S: Waring laboratory Science, Torrington, CT, USA), and the pooled solutions were diluted to a 1% (*w*/*v*) stock using 0.1 M potassium phosphate buffer. This stock suspension was stored at 4 °C and diluted to the desired concentration prior to use. Bacterial isolates were streaked onto 0.4% chitin nutrient agar (CNA). Isolates which produced a clearing zone were characterized as positive for chitin hydrolysis [22].

#### 2.2.2. Dual-Culture Assay

An in vitro plate assay was used to screen isolates for the ability to reduce growth of two apple pathogens: *Venturia inaequalis* and *Colletotrichum acutatum*. A single-spore culture of *V. inaequalis* RSGD1.1.1 was obtained from Dr. M. Jimenez Gasco at Penn State University [23]. *C. acutatum* was isolated from infected fruit at the PSU FREC [24]. A mycelial plug of each pathogen was placed on potato dextrose agar (PDA) in a 100 mm Petri dish. A streak of the candidate bacterial isolate was placed 5 cm from the mycelial plug. Control plates consisted of the mycelial plug without bacteria. Three replicate plates were prepared for each pathogen–bacterial isolate combination. Plates were incubated at 20 °C, and radial growth was measured every two days for ten days. The final colony diameter perpendicular to the bacterium at the end of the experiment was also determined. Data were analyzed for statistical significance by PROC MIXED followed by a Dunnett’s test using SAS (SAS Institute, version 9.2; Cary, NC, USA) to determine if the bacteria were able to significantly suppress fungal growth compared to the control. 

#### 2.2.3. In Vivo Antifungal Assay

Preliminary experiments were conducted to screen isolates for the ability to reduce severity of bitter rot disease (caused by *C. acutatum*) in vivo. The candidate bacterial isolates were grown in 100 mL of sterile tryptic soy broth in 250 mL Erlenmeyer flasks. Flasks were incubated for seven days at 28 °C and 120 rpm on a rotary incubator shaker (New Brunswick Scientific Model M1024-000, Edison, NJ, USA). Bacterial cells were harvested by centrifugation at 3800 rpm for 15 min at 4 °C using a Sorvall RT7 centrifuge (ThermoFisher Scientific, Waltham, MA, USA). The supernatant was discarded, and the bacterial pellets were re-suspended in 0.1 M potassium phosphate buffer and adjusted to a concentration of 1 × 10^7^ CFU/mL. *C. acutatum* was prepared as described by Poleatewich et al. [25]. Apple cv. Rome Beauty fruit were harvested from trees planted in an experimental orchard at the PSU FREC and washed with a 5% commercial bleach solution and rinsed with tap water. All fruit were wounded at the equator to a depth of 4 mm using a sterilized six-penny nail. Nails were mounted through a rubber stopper to ensure uniformity of wound depth. Immediately after wounding, the wounds of 50 fruit per isolate were inoculated with 20 µL of the candidate biocontrol bacterial isolate or a sterile distilled water control. One hour after bacterial application, wounds of 25 fruit per treatment were inoculated with 20 µL of a 4.0 × 10^4^ conidia/mL (800 spores/wound) spore suspension of *C. acutatum* as described by Poleatewich et al. [26]. The remaining 25 fruit per treatment (not inoculated with *C. acutatum*) served as controls to determine if the bacteria alone could cause symptoms on the fruit. Once the inoculum dried, the fruits were randomly placed on 20-count molded cardboard trays. The trays were stored in a dark walk-in growth room at 22 °C on plastic shelves. Fruits were evaluated for symptom severity seven days after pathogen challenge. To assess disease severity, the lesion diameter was measured laterally and horizontally across the wound site, and the total lesion area was estimated using the average of the two measurements. Data were analyzed for statistical significance by PROC MIXED followed by a Dunnett’s test using SAS (SAS Institute, version 9.2; Cary, NC, USA) to determine if the bacteria were able to significantly suppress lesion size compared to the water control.

### 2.3. Evaluation of Isolates for Disease Suppression on Fruit

Based on the results from the preliminary in vitro and in vivo experiments, nine isolates were selected for further evaluation. Bacteria and pathogen cultures were grown as described above. Fruits were harvested from cv. Golden Delicious and Rome Beauty trees grafted on M.26 rootstock planted at the PSU FREC. Experiments were conducted on symptom-free Golden Delicious and Rome Beauty fruit in separate experiments. The experiment consisted of a 10 × 2 factorial with ten bacterial treatments (nine isolates + water control) and two pathogens. Each of the twenty treatments were applied to ten replicate fruits. For both cultivars, the fruits were prepared, wounded, and inoculated with the bacterial isolates as described above. One hour after bacterial application, wounds were inoculated with 20 µL of a 4.0 × 10^4^ conidia/mL (800 spores/wound) spore suspension of *C. acutatum* or *P. expansum* as described by Poleatewich [26]. Once the inoculum dried, the fruit were placed in ten 20-count plastic trays and covered with a plastic lid (containing five holes for air circulation). The trays were stored in a dark walk-in growth room at 20 °C and ~24% relative humidity on plastic shelves. Fruits were evaluated for symptom severity seven days after pathogen challenge. To assess disease severity, the lesion diameter was measured laterally and horizontally across the wound site, and the total lesion area was estimated using the average of the two measurements. Data were analyzed for significance using the mixed procedure of SAS 9.2 with tray as the random variable followed by a Tukey test (α = 0.05) to identify differences in mean lesion areas across treatments.

### 2.4. Identificaion of Biocontrol Isolates

Select bacterial isolates (chosen based on efficacy in preliminary screening assays) were streaked onto fresh YED agar one day before PCR amplification. This short growth period was used to ensure the bacterial cells were in the vegetative state. A small colony was removed from the surface of the YED plate with a 10 µL pipette tip and placed into a PCR tube containing 20 µL of master mix. The master mix contained 2 µL of 10× PCR buffer with 1.5 mM MgCl, 1.6 µL dNTP mic (200 µM each), 0.4 µL 530f primer (910 µM), 0.4 µL 1392 r primer (10 µM), and 0.2 µL Taq Polymerase (Gene Choice, San Diego, CA, USA) and consisted of a total volume of 20 µL. The universal primers used to amplify the small-subunit ribosomal RNA were 530f (5′-GTGCCAGCMGCCGCGG) and 1392 r (5′-ACGGGCGGTGTGTRC). PCR amplification was conducted using an Eppendorf Mastercycler Personal Thermal Cycler (Eppendorf AG, Hamburg, Germany) with the following cycle: 5 min at 95 °C followed by 35 cycles of 94 °C for 15 s, 72 °C for 15 s, and a final extension at 72 °C for 5 min. PCR products were cleaned using ExoSAP-IT (USB Corp., Santa Clara, CA, USA).

Sanger sequencing of PCR products was performed using an ABI Hitachi 3730 XL DNA Analyzer (Hitchi Ltd., Tokyo, Japan) at the Penn State Genomics Core Facility. Sequences were aligned using MAFFT alignment software version 7.305 b with default settings [27]. The resulting alignment was trimmed using trimAL version 1.4rev15 with a gap threshold of 0.5 [28]. Sequences were analyzed using the National Center for Biotechnology Information (NBCI) Basic Local Alignment Search Tool (BLAST) search algorithm. The identities of the five apple isolates were determined by comparison with high-scored rRNA sequences in BLAST searches and sequences from *Bacillus*, *Brevibacillus,* and *Paenibacillus* type strains. The 16S rRNA sequence data from the five apple isolates and 37 strains were used to create phylogenetic trees. Phylogenetic trees were inferred and compared using IQTree (version 1.6.12, default settings) bootstrapping methods ultrafast bootstrap (UFBoot) and SH-aLRT test [29]. 

### 2.5. Colonization of Fruit Wounds by Biocontrol Isolates

A postharvest experiment was conducted to assess bacterial colonization of fruit wounds at room temperature and in cold storage. This experiment consisted of a 5 × 2 factorial with five bacterial isolates (A1-1, A3-6, Ae-1, FLS-1, and FLS-5) and two temperatures (20 °C and 2 °C) for a total of ten treatments. The experiment was conducted on cv. Golden Delicious fruit and repeated on cv. Rome Beauty fruit. Fruits were wounded to create two wounds per fruit, and 20 µL of 10^7^ CFU/mL of bacteria was added to each wound on sixteen fruit per bacterial isolate (2.0 × 10^5^ CFU/wound). Eight fruit per isolate treatment were placed in plastic trays with lids and stored at 20 °C, and eight fruit of each isolate treatment were stored in a walk-in cold room at 2 °C. Wound colonization of fruit stored at 20 °C was determined at one-, two-, four- and eight-days post inoculation (dpi) and one, seven, fourteen, and twenty-eight dpi for fruit stored at 2 °C. To assess colonization, wound tissues from two replicate fruit were excised using a #4 cork borer (44.2 mm^2^) to a depth of approximately 4 mm. A single core was placed in a 101 mm × 152 mm stomacher filter bag with 5 mL of 0.1 M potassium phosphate buffer and triturated in a stomacher blender (Interscience, MiniMix, Saint Nom la Bretêche, France) for 30 s. 50 µL of the undiluted and 10-fold diluted triturate was plated in duplicate on yeast extract dextrose agar (YED). Plates were incubated at 20 °C for 24 h and enumerated. The minimum detectable population for this methodology was log 2.0 CFU/wound. 

### 2.6. Cytological Investigation of Disease Suppression

Scanning electron microscopy (SEM) was used to observe interactions between the biocontrol bacteria and the postharvest pathogens, *C. acutatum* and *P. expansum*, in fruit wounds. This experiment consisted of a 5 × 2 × 2 factorial with five bacteria treatments (A3-6, Ae-1, FO-20, A3-2, and none), two pathogens (*C. acutatum* and *P. expansum*), and two sampling times (24 h and 4 dpi). The five bacterial isolates were chosen to represent positive and negative controls with respect to suppression of lesion size caused by *C. acutatum* and *P. expansum*. A sterilized six-penny nail was used to make four wounds on each Golden Delicious fruit. Wounds were then inoculated with 20 µL of the respective bacterial isolate (sterile water for no-bacteria controls) at a concentration of 10^7^ CFU/mL. One hour after bacterial application, wounds were challenged with 20 µL of 3.0 × 10^4^ conidia/mL suspension of either *C. acutatum* or *P. expansum*. A hemacytometer (American Optical, Buffalo, NY, USA) was used to adjust the concentration of the pathogen inoculum prepared as described earlier. Fruit was placed in 20-count molded plastic trays and placed in a dark walk-in growth room at 20 °C. 

A 10 mL stock fixing solution was prepared with 5 mL of 0.2 M potassium phosphate buffer (pH 7.4), 1.6 mL of 16% paraformaldehyde, 0.6 mL of 25% glutaraldehyde, and sterile distilled water added to a final volume of 10 mL. The final concentration of the fixing solution was 25% paraformaldehyde and 1.5% gultaraldehyde. One ml of fixing solution was pipetted into each of ten vials. Wound tissue was excised using a sterilized #4 core borer (44.2 mm^2^) to a depth of approximately 5 mm. Tissue samples were collected 24 h and 4 dpi and immediately immersed in the fixing solution for 24 h at 4 °C. Four tissue samples per treatment were placed in each vial (containing 1 mL of fixative). Following the 24-h fixation, samples were washed with 0.2 M potassium phosphate buffer (pH 7.4) in three five-minute wash cycles. After washing, samples were dehydrated in a gradient ethanol series at room temperature. Next, samples were critical-point dried using a Bal-Tec CPD030 critical point dryer (Techno Trade, Manchester, NH, USA). The dried samples were immediately mounted on aluminum stubs (15 mm × 10 mm) using double-sided carbon tape and stored in a desiccator cabinet. The next-day samples were sputter coated with 10 nm of gold/palladium (2 min at 10 mAmps) using a Bal-Tec SCD-050 sputter coater. Following sample preparation, tissues were viewed using a JOEL JSM 5400 scanning electron microscope (JOEL; Peabody, MA, USA) at 10 kV, 0-degree tilt, and a 20 mm working distance.

## 3. Results

### 3.1. Isolation and Preliminary Screening of Bacterial Isolates from Apple

A total of 75 endospore-forming bacterial isolates were collected from feral trees and conventional and abandoned orchards in 2006 and 2007. Twenty-six isolates (35%) were positive for chitin hydrolysis as indicated by the production of a clearing zone surrounding the colony on 0.4% CNA. Eleven isolates (Ae-1, A1-1, A3-6, A3-2, A3-3, FLS-1, FLS-5, FO-1, WGD-5, WS-1, and WS-3) significantly (*p* < 0.001) reduced radial growth of *C. acutatum* hyphae on PDA (Table 1). Isolates A1-1, FLS-5, FO-1, and WS-1 resulted in the greatest suppression in radial growth of *C. acutatum* compared to no-bacteria control plates, ranging from 41% to 64% reduction in total colony size after the ten-day experiment. Isolates A1-1, A1-11, FO-1, and A3-F1 significantly (*p* < 0.002) reduced radial growth of *V. inaequalis* on PDA. Isolates A1-1 and FO-1 significantly reduced the growth of both pathogens assayed in vitro (Table 1). Six isolates (A1-1, A3-2, A3-3, A3-6, A3-F1, Ae-1, and FLS-1) reduced both growth in vitro and symptom severity in vivo of *C. acutatum* (Table 1).

### 3.2. Suppression of C. acutatum and P. expansum in Fruit Wounds

Two isolates tested (A3-6 and Ae-1) resulted in significant suppression of bitter rot lesion size on cv. Golden Delicious (*p* < 0.001) and Rome Beauty (*p* < 0.01) fruit, and one isolate (A3-2) suppressed blue mold lesion size on cv. Golden Delicious fruit (*p* = 0.049) (Figure 1). None of the isolates tested resulted in significant suppression of lesion size caused by both pathogens.

### 3.3. Isolate Identification

Using phylogenetic analyses, it was determined that isolates A1-1, A3-6, Ae-1, and FLS-5 belong to the genus *Bacillus,* and isolate FLS-1 belong to the genus *Brevibacillus* (Figure 2). Isolate A1-1 and FLS-5 were determined to be members of the *B. cereus* group. Isolate FLS-5 was most similar to *B. cereus*. Isolate A1-1 was most similar to *B. mycoides* and *B. weihenstephanensis*. Based on the mycoidal colony morphology on nutrient media, isolate A1-1 was concluded to be *B. mycoides*. Isolates A3-6 and Ae-1 were most similar to *B. megaterium*. Sequence data from the five apple isolates were submitted to GenBank (accession IDs OP596281-88).

### 3.4. Colonization of Fruit Wounds

All the bacterial isolates tested were able to colonize wounds on cv. Golden Delicious and Rome Beauty fruit at 20 °C and 2 °C (Figure 3). Approximately 2.0 × 10^5^ CFU/mL were applied to each wound. Colonization levels by isolates FLS-5 and A1-1 were similar at the two temperature treatments. Isolate FLS-5 averaged between log 5.3 and log 6.8 CFU/wound at 20 °C and between log 5.3 and log 6.3 CFU/wound at 2 °C. Colonization by isolates A3-6 and Ae-1 was greater at 20 °C compared to 2 °C on both cultivars tested, ranging between log 3.6 and log 5.0 at 20 °C and between log 2.2 and log 3.8 CFU/wound at 2 °C.

### 3.5. Cytological Investigation of Disease Suppression

Suppression of bitter rot and blue mold symptoms by application of the bacterial isolates into wounds of cv. Golden Delicious fruit was investigated by SEM. Isolates A3-6 and Ae-1 were chosen for their ability to suppress *C. acutatum*, and isolate A3-2 was chosen for its ability to suppress *P. expansum*. Isolate FO-20 has not shown significant suppression of either pathogen in previous experiments and was thus chosen as a negative control. Observations of fruit inoculated in parallel with SEM observations confirmed isolates A3-6 and Ae-1 inoculated fruit developed lesions that were smaller than the control (*p* < 0.05), and FO-20 and A3-2 had no effect on lesion size caused by *C. acutatum* (Appendix A). Observations of wounded tissue colonized with isolate A3-6 showed attachment of bacterial cells to the hyphae of *C. acutatum* in samples taken at 4 dpi (Figure 4). Closer examination revealed sections of the hyphae where bacterial cells had attached were damaged and collapsed (Figure 4), while sections with no attached bacteria appeared normal. Wound tissue samples inoculated with isolate Ae-1 and challenged with *C. acutatum*, appeared to have higher numbers of bacterial cells covering the wound site at 4 dpi (Figure 4). Closer examination indicated Ae-1 did not attach or cause damage to hyphae of *C. acutatum* (Figure 4).

For blue mold, inoculations confirmed fruit treated with isolate A3-2 had significantly smaller lesions compared to the control (Appendix A). Under SEM, none of the isolates attached or appeared to cause any physical damage to hyphae of *P. expansum* in apple wounds (Figure 5). Conidiophores bearing conidia were only observed in water control wounds.

## 4. Discussion

Several bacterial isolates in this study exhibited biocontrol potential in vitro and reduced severity of bitter rot and blue mold when applied to wounded apple fruit postharvest. Isolates A3-6 and Ae-1 resulted in the greatest suppression of bitter rot lesion area. Analyses indicate isolates Ae-1 and A3-6 were most similar to *B. megaterium.* The species *B. megaterium* has been shown to have biocontrol activity, including reduction in postharvest decay of peanut kernels caused by *Aspergillus flavus* [30] and *Septoria tritici* blotch of wheat [31]. Interestingly, *B. megaterium* isolates have also shown potential to inhibit growth of the human pathogen, *Listeria monocytogenes* [32]. In our previous research, isolate A3-6 also suppressed apple scab on leaves and fruit and bitter rot on apple fruit in storage [24]. While fruit treated with *B. megaterium* isolates A3-6 and Ae-1 had the smallest bitter rot lesions in this study, wound colonization levels were typically log 2 lower than *B. laterosporus* isolate FLS-5 at 2 °C and log 1 lower at 20 °C. These observations suggest the mechanism of bitter rot disease suppression by isolates A3-6 and Ae-1 may not involve competition for space or nutrients. This hypothesis was substantiated for isolate A3-6 in the SEM experiment in which hyphal tissues of *C. acutatum* had collapsed where the bacterial isolate had attached to the hyphae suggesting parasitism as a mode of action. In contrast, isolate Ae-1, which also suppressed bitter rot lesion size on fruit, did not appear to attach or damage hyphal tissues of *C. acutatum*. These observations suggest the mode of action for isolate Ae-1 is different from isolate A3-6 even though both isolates reduced fungal growth in the plate assays and did not produce a clearing zone on chitin-amended media, indicating the isolates do not produce chitinase. Bertagnolli et al. [33] identified several extracellular enzymes produced by *B. megaterium* strain B153-2-2, including protease and pectin lyase, capable of reducing growth of *Rhizoctonia solani* in vitro. In agreement with our study, Bertagnolli also indicated a lack of detectable chitinase activity by *B. megaterium*. Additional experimentation is needed to further elucidate the mechanism of disease suppression by *B. megaterium* isolate A3-6 and Ae-1, including characterization of secondary metabolites and their potential role in suppression of pathogen growth.

In this study, isolate *B. mycoides* A1-1 and *B. cereus* FLS-5 were positive for chitin hydrolysis and reduced growth of *C. acutatum* in vitro. In our previous research, A1-1 and FLS-5 significantly reduced apple scab severity on leaves and fruit and bitter rot in storage [24]. However, in this study isolate A1-1 did not consistently suppress bitter rot symptoms, and isolate FLS-5 was not effective in any in vivo experiments despite maintaining the highest population level in fruit wounds. Analyses indicate isolate A1-1 most closely resembles *B. mycoides* and *B. weihenstephanensis*. Mycoidal growth of isolate A1-1 on nutrient media allowed us to place this isolate in the sub-group *B. mycoides* as *B. weihenstepanesis* has a non-mycoidal colony morphology [34]. Phylogenetic analyses indicate isolate FLS-5 belongs to the *B. cereus* group. *B. cereus* is found in diverse habitats, is ubiquitous in soil [35], and can have biocontrol activity. However, some members of this group can cause two types of food poisoning, a diarrheal type and an emetic type, which are caused by different toxins [36]. Some *B. cereus* strains however, lack enterotoxin genes [37], and several have demonstrated biocontrol activity on several crops [21,38,39] and many are chitinolytic [38,40].

Isolate FLS-1 was positive for chitin hydrolysis, reduced growth of *C. acutatum* in vitro, and suppressed bitter rot symptoms in preliminary screenings but did not suppress symptoms in follow up experiments. Isolate FLS-1 was identified as *Brevibacillus laterosporus*, an aerobic spore-forming bacterium that has documented biocidal activity towards insects [41].

Several isolates in this study reduced pathogen growth in vitro but were not effective in vivo (Table 1). Similar observations have been reported in the literature as in vitro assays; while quick and inexpensive, are generally poor predictors of field performance [42] in part because of the artificial growth conditions compared to in vivo studies [19,43]. For example, Burr et al. [44] found no correlation between in vitro antibiosis and the ability of bacteria and yeasts to suppress apple scab on seedlings. Additionally, the selection of organisms solely based on in vitro tests (also known as dual-culture assays) may become problematic as potentially useful organisms, whose mechanism of disease suppression as anything other than antibiosis or direct parasitism will be overlooked. For example, the use of in vitro assays for screening may fail to identify inducers of induced systemic resistance (ISR) and systemic acquired resistance (SAR) [19,45].

In this study, we utilized chitin hydrolysis as a BCA-screening strategy. Many studies have focused on selecting and testing antagonistic bacteria, specifically those with chitinolytic properties for use as BCAs [21,40,46]. Assays to screen isolates for chitin hydrolysis have relied on conventional plate methods in which colloidal chitin is incorporated into an agar base [22]. Several species of the genus *Bacillus* have been shown to produce chitinase enzymes, including *B. circulans* [47], *B. licheniformis* [48], *B. macerans* [49], and *B. cereus* [38]. Because chitin is a major constituent of fungal cell walls, bacterial chitinase enzymes are thought to be useful for biocontrol of fungal plant pathogens [50], and several chitinolytic microorganisms have demonstrated biocontrol activity against fungal plant pathogens [40,51,52]. In this study, many of the isolates that were positive for chitinase production also demonstrated biocontrol potential (either via inhibition of pathogen growth or suppression of symptoms), although chitinase production was not necessary for biocontrol activity.

This research demonstrates apple orchards can be a source of new BCAs for control of postharvest diseases of apple. Furthermore, abandoned orchards that have returned to their “wild state” may serve as a resource to identify novel BCAs or consortia of biocontrol microbes. Of the top seventeen isolates exhibiting biocontrol potential in vitro or in vivo, twelve (70%) were collected from abandoned orchards or natural (unmanaged) locations. We also noticed, anecdotally, abandoned orchards were replete with healthy trees. These observations suggest abandoned orchards may contain unique microbial communities that are important for plant health, but further research is needed to understand these interactions. It is known that agricultural management and the persistent use of chemical pesticides and fertilizers in agroecosystems can alter plant-associated microbial communities [20,53,54]. There are limited examples however, of research comparing microbial community composition in abandoned and managed systems. Several studies have utilized abandoned agricultural land to study the effect of management practices on insect pests and their natural predators. For example, Altieri and Schmidt [55] found a negative correlation between plant-beneficial predator insect abundance and intensity of management practices. The rate of predation by beneficial species was highest in the abandoned orchard and the organic orchard planted with a cover crop. Furthermore, natural enemies tended to establish long term in organic orchards compared to conventional orchards. Taken together, this literature provides evidence that abandoned agroecosystems may harbor disease suppressive consortia of microbes. Further research is needed to test these hypotheses and identify implications for biocontrol in tree fruit production.

## 5. Conclusions

In conclusion, several isolates were identified in this research that suppress growth of *C. acutatum*, *P. expansum*, and *V. inaequalis* in vitro and reduced lesion size on apple fruit caused by *C. acutatum* and *P. expansum*. Future studies are needed to characterize these isolates, determine application rates, and determine compatibility with orchard IPM practices. This research also demonstrates abandoned apple orchards can be a source of new BCAs for suppression of postharvest diseases of apple.

## Figures and Tables

**Figure 1 microorganisms-11-00081-f001:**
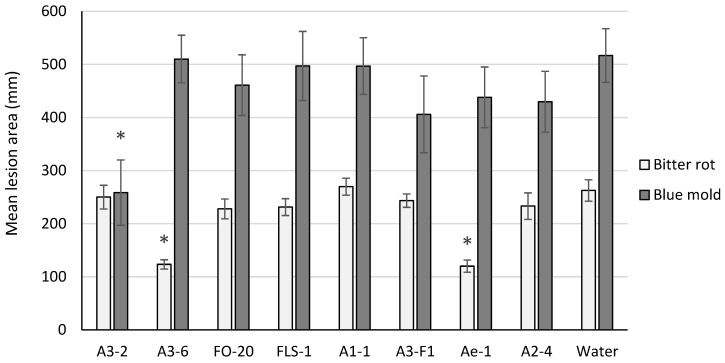
Mean lesion area on cv. Golden Delicious fruit six days post inoculation with *C. acutatum* (bitter rot) or *P. expansum* (blue mold). For each pathogen tested, bars with an asterisk (*) were significantly different from the water control at α = 0.05. Error bars represent the standard error of the mean.

**Figure 2 microorganisms-11-00081-f002:**
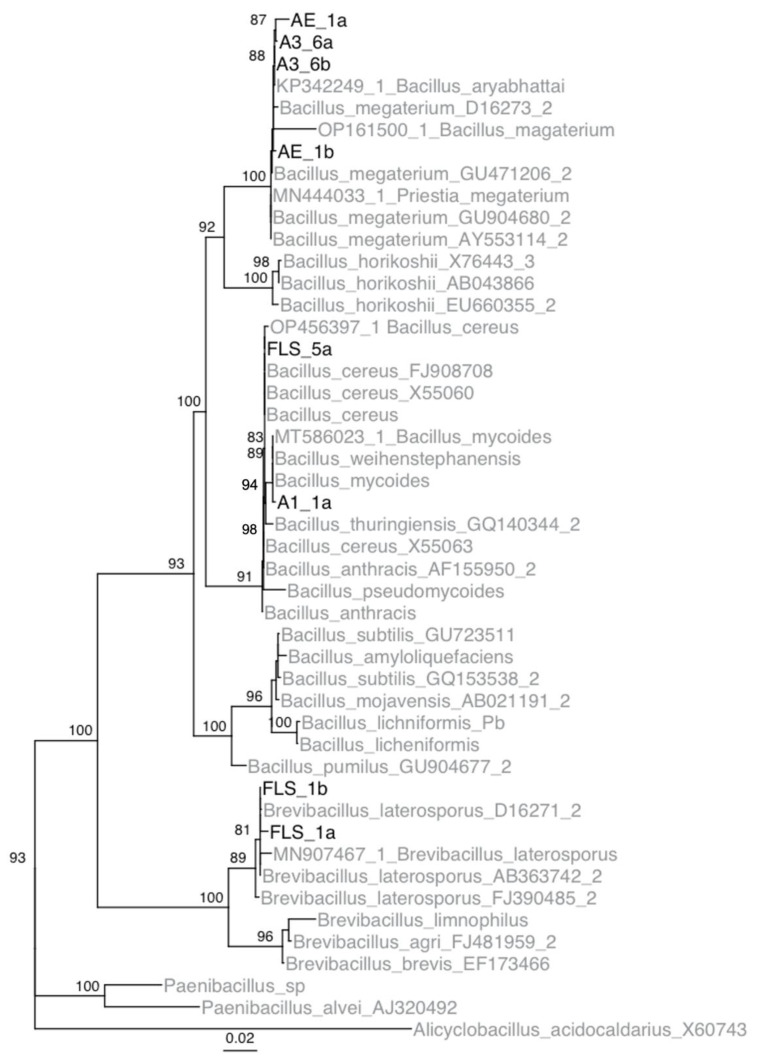
Inferred IQTree ultrafast bootstrap (UFBoot) of the isolates collected from Pennsylvania and type strains of *Bacillus* species based on 16S rDNA sequences. Numbers located at each node are bootstrap values. Note, bootstrap values below 80 are not shown on the tree. Isolates from this study are shown in black, and isolates obtained from GenBank are in gray.

**Figure 3 microorganisms-11-00081-f003:**
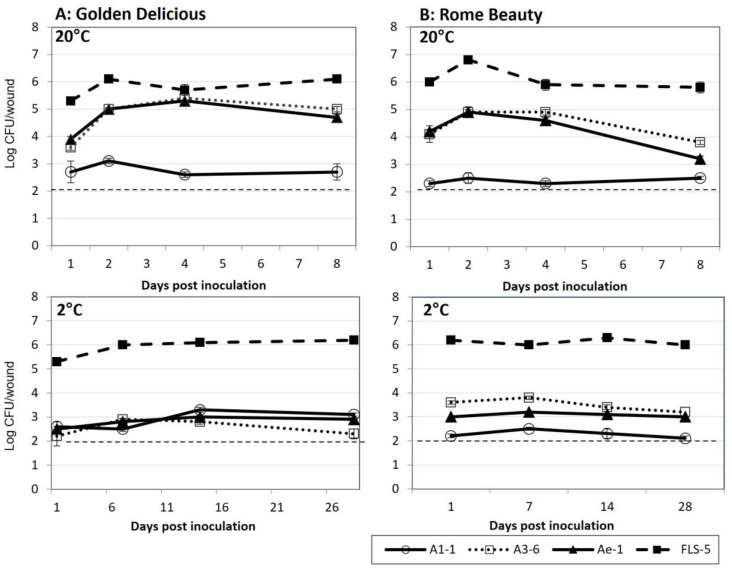
Colonization of fruit wounds by bacterial isolates A1-1, A3-6, Ae-1, and FLS-5 at 20 °C and 2 °C on (**A**) Golden Delicious and (**B**) Rome Beauty. Data are reported as log CFU/wound. Wounds were inoculated with 2.0 × 10^5^ CFU/wound. Error bars represent the standard error of the mean. The dashed line indicates the minimum detection level of the experiment.

**Figure 4 microorganisms-11-00081-f004:**
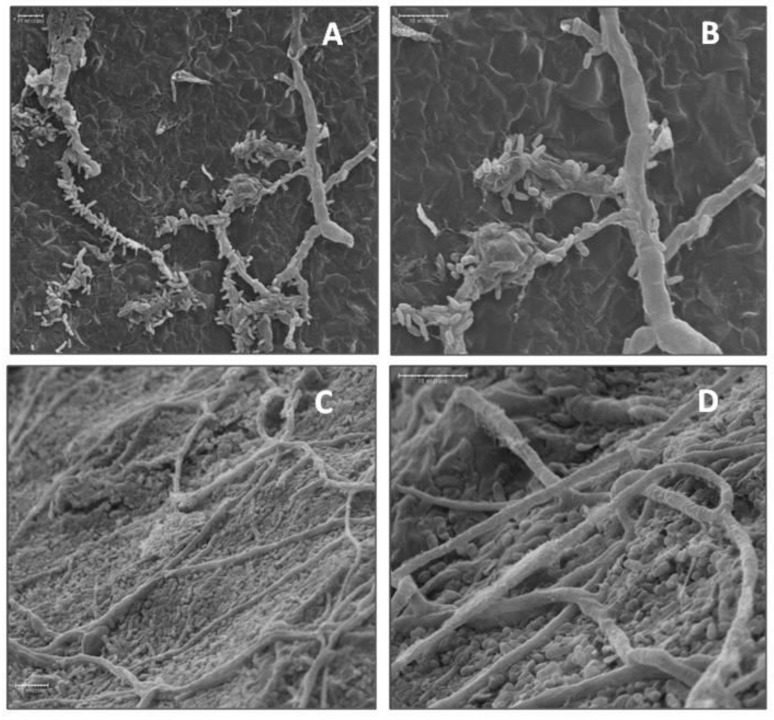
Scanning electron micrographs of biocontrol bacterial cells interacting with hyphae of *C. acutatum* in apple wounds four days after inoculation. (**A**) Attachment of biocontrol isolate A3-6 cells to hyphae of *C. acutatum* at 750×. (**B**) Attachment and degradation of *C. acutatum* hyphae by isolate A3-6 at 1500×. (**C**) Isolate Ae-1 in fruit wounds at 1000× and (**D**) 2000×.

**Figure 5 microorganisms-11-00081-f005:**
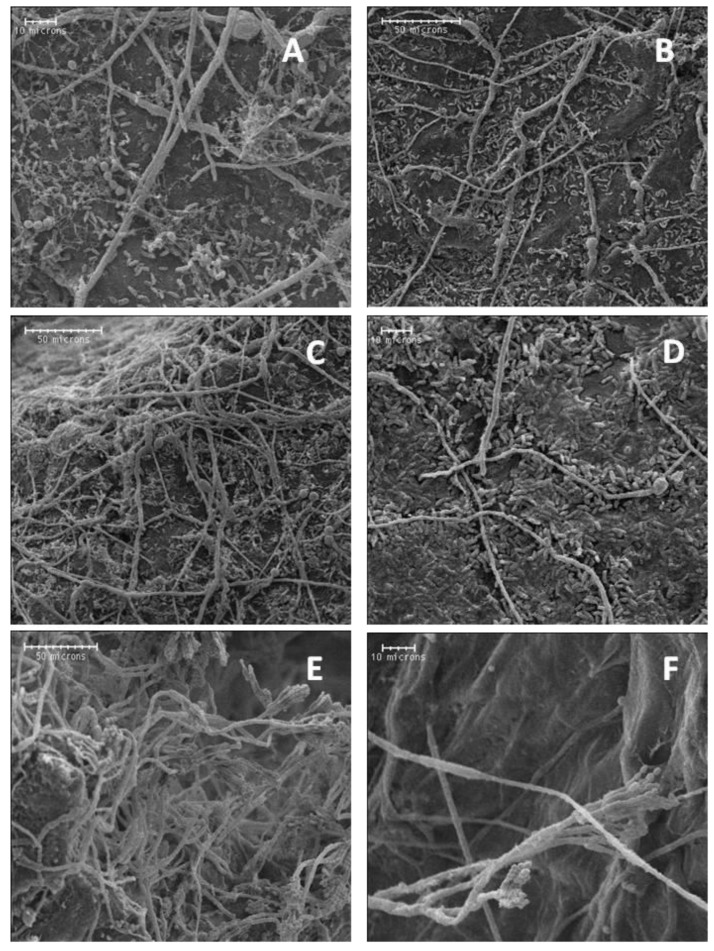
Scanning electron micrographs of biocontrol bacterial cells interacting with hyphae of *P. expansum* in apple wounds four days after inoculation. (**A**,**B**) Isolate A3-6 at 500× and 1000×, (**C**,**D**) isolate A3-2 at 500× and 1000×, and (**E**,**F**) no-bacteria control at 500× and 1000×.

**Table 1 microorganisms-11-00081-t001:** Results from preliminary in vitro and in vivo screening of bacterial isolates collected from apples in Pennsylvania. Shaded isolates suppressed growth both in vitro and in vivo.

		In Vitro Plate Assay			In Vivo Assay ^z^
Isolate ^v^	Location ^w^	Chitinase ^x^	*C. acutatum* ^y^	*V. inaequalis*	*C. acutatum*
A1-1	abandoned	+	+	+	*
A1-11	abandoned	−	−	+	-
A2-4	abandoned	+	−	−	*
A3-2	abandoned	+	+	−	**
A3-3	abandoned	+	+	−	**
A3-4	abandoned	+	−	−	*
A3-6	abandoned	−	+	−	**
A3-F1	abandoned	+	−	+	**
Ae-1	abandoned	−	+	−	**
FC-2	FREC no spray	+	−	−	-
FLS-1	FREC Reduced risk	+	+	−	*
FLS-5	FREC Reduced risk	+	+	−	-
FO-1	FREC Organic	+	+	+	-
FO-20	FREC Organic	+	−	−	*
WGD-5	natural	+	+	−	-
WS-1	natural	−	+	−	-
WS-3	natural	+	+	−	-

^v^ Results are shown for isolates that suppressed growth in the in vitro or in vivo experiments. ^w^ Isolates were collected from abandoned orchards, feral trees in forested natural areas, and research orchards at the Penn State Fruit Research and Extension Center (PSU FREC). ^x^ Isolates marked as + produced a clearing zone on 0.4% chitin nutrient agar. Isolates marked as − did not produce a clearing zone. **^y^** Isolates marked as + significantly reduced pathogen lesion size (α = 0.05) compared to the control. Isolates marked as − were not significantly different from the control. ^z^ Isolates were significantly different from the water control at α = 0.05 (*) or α = 0.01 (**).

## Data Availability

Data available upon request. Sequence data of identified isolates were deposited to GenBank under accession IDs OP596281-88.

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
