# Peer review of "Evaluation of Endospore-Forming Bacteria for Suppression of Postharvest Decay of Apple Fruit"

_microorganisms, 2022, doi:10.3390/microorganisms11010081_

Round 1

Reviewer 1 Report

Dear Authors,

Your research paper entitled 'Evaluation of endospore-forming bacteria for suppression of postharvest decay of apple fruit' focused on an interesting and important issue seeking bio-control regent(s) to protect apple fruits from postharvest decay caused by various pathogens. The current manuscript includes many interesting results that can lead to reasonable conclusions. However, the 'Conclusions' section in the present form (line 446-450) seems not OK due to many over-evaluation perspectives, so please modify or rewrite this section with clear conclusions directly related with the present main results. Also, there are many unclear or confused word or phrases through the present manuscript. For examples, 'bitter rot and blue mold apple' (line 14), 'natural areas' (line 15), 'bitter rot symptom severity' (line 19), 'severity of blue mold disease severity' (line 20, how to define 'severity' or how to evaluate severity degree?), and so on. Not only in the Abstract section but also in the following main sections contained many improper phrases like those and all need revisions. In addition, just my suggestion, it is better to provide some phenotype pictures (apple fruit pictures) as representatives with the results of Figure 1 and Figure 2.

Author Response

Response to Reviewer 1 comments

Point 1: Your research paper entitled 'Evaluation of endospore-forming bacteria for suppression of postharvest decay of apple fruit' focused on an interesting and important issue seeking bio-control regent(s) to protect apple fruits from postharvest decay caused by various pathogens. The current manuscript includes many interesting results that can lead to reasonable conclusions. However, the 'Conclusions' section in the present form (line 446-450) seems not OK due to many over-evaluation perspectives, so please modify or rewrite this section with clear conclusions directly related with the present main results.

Thank you for the suggestion we have revised the conclusions section to more concisely discuss conclusions related to the research results.

Point 2: Also, there are many unclear or confused word or phrases through the present manuscript. For examples, 'bitter rot and blue mold apple' (line 14), 'natural areas' (line 15), 'bitter rot symptom severity' (line 19), 'severity of blue mold disease severity' (line 20, how to define 'severity' or how to evaluate severity degree?), and so on. Not only in the Abstract section but also in the following main sections contained many improper phrases like those and all need revisions.

We have carefully reviewed the manuscript to correct any errors and unclear phrases. On lines 170-173 we explain that disease severity was measured as lesion size on apple fruit. We edited the text in several places to clarify this point. We evaluated degree of severity as size of the lesion. Lesion area (sq mm) data were analyzed to determine if fruit treated with bacteria has smaller lesions compared to the control.

Point 3: In addition, just my suggestion, it is better to provide some phenotype pictures (apple fruit pictures) as representatives with the results of Figure 1 and Figure 2.

This is a great suggestion. Unfortunately, we do not have representative pictures from this study. In a follow-up manuscript (under preparation) we do have pictures of the results.

Reviewer 2 Report

The paper is well-written, contains clear and relevant information regarding the microbial biocontrol agents for control of postharvest disease. The introduction is brief and provides sufficient information on the importance of choosing the topic of this scientific paper.

The methods are generally appropriate.  At this part I have a few suggestions:

Line 143…".. was prepared as described by Poleatewich et al (Poleatewich et al. 2010)." Should be use the number of the reference, not the name of Author, twice.

Line 174 ……the same comment like before, the reference (Poleatewich et al. 2010) is presented as the name of author and year, but in the first part of the manuscript, the references were presented with a number.

This work presents interesting results that are clear and compelling; the authors making an important contribution to the research literature in this area of investigation.

Author Response

Point 1 and 2: Line 143…".. was prepared as described by Poleatewich et al (Poleatewich et al. 2010)." Should be use the number of the reference, not the name of Author, twice. Line 174 ……the same comment like before, the reference (Poleatewich et al. 2010) is presented as the name of author and year, but in the first part of the manuscript, the references were presented with a number.

The citations have been corrected. Thank you for the feedback.

Reviewer 3 Report

The paper describes a study on endospore-forming bacteria possibly useful as a biocontrol agent against postharvest fruit decay of apple. The manuscript is reasonably well-written and it presents some interesting data but it should be carefully revised before it can be processed further. Some typos (check species names!) and grammar errors can also be found and should be corrected. Specific comments and suggestions were given below.

Introduction is slightly oversized. There is plenty of detailed information that is not directly linked to the subject of the study and could easily be removed without any damage to the story. This applies to the list of registered biocontrol commercial products, a list of references would be sufficient.

Materials and Methods: Section 2.1 heading should be expanded as also isolation and culture conditions were included and there is no separate section for that part of the procedure. Also, I believe that only leaf tissue handling was described and what about fruit samples? Section 2.2 should be divided into subsections (chitin degradation assay, dual culture tests, in vivo antifungal activity) to make it more clear and reader-friendly. It is confusing that three pathogenic species are mentioned (P. expansum, C. acutatum and V. inaequalis), while from the context only two are expected. I understand that they were used in different experiments, but this should be better explained. The section devoted to identification of the selected bacterial strains should be moved up. I would put it before Section 2.3. Furthermore, used pathogens should also be characterized or referenced to previous works. Lines 208-211 seem like results and should be moved there or re-written. Finally, why only five sequences of bacterial strains were analysed and not all nine of them?

Results are presented in a clear and convincing manner. Concerning phylogenetic analysis, the sequences of tested strains should be provided (possibly in Supplementary materials) or at least deposited in GenBank database to show Accession Numbers on the dendrogram. Otherwise, it is impossible to verify this analysis.

Discussion is focused and I do not have major complaints concerning this section.

Author Response

Point 1: Introduction is slightly oversized. There is plenty of detailed information that is not directly linked to the subject of the study and could easily be removed without any damage to the story. This applies to the list of registered biocontrol commercial products, a list of references would be sufficient.

We agree with this feedback and have revised the introduction to focus on information directly related to the objectives of the research.

Point 2: Materials and Methods: Section 2.1 heading should be expanded as also isolation and culture conditions were included and there is no separate section for that part of the procedure. Also, I believe that only leaf tissue handling was described and what about fruit samples?

We revised the heading for section 2.1 to ‘isolation and culture of biocontrol bacteria” to better describe the content in the section. Methods for sampling fruit tissue were added on lines 111-112.

Point 3: Section 2.2 should be divided into subsections (chitin degradation assay, dual culture tests, in vivo antifungal activity) to make it more clear and reader-friendly. It is confusing that three pathogenic species are mentioned (P. expansum, C. acutatum and V. inaequalis), while from the context only two are expected. I understand that they were used in different experiments, but this should be better explained.

This is a great suggestion. We have added three subsections for each of the three assays described in this section. Preliminary screening in vitro and in-vivo assays were conducted using V. inaequalis and C. acutatum. Based on the results of these preliminary experiments, elite isolates were evaluated for suppression of C. acutatum and P. expansum.

Point 4: The section devoted to identification of the selected bacterial strains should be moved up. I would put it before Section 2.3.

We moved this section as suggested. We also moved reporting of the results to keep order of topics presented in the methods and results consistent.

Point 5: Furthermore, used pathogens should also be characterized or referenced to previous works.

Citations were added to lines 137-139

Point 6: Lines 208-211 seem like results and should be moved there or re-written.

Yes, this text describes results as context for the methods in this section. Our strategy was to provide this information to explain why these specific isolates were chosen for the experiment. The goal of this experiment was to compare isolates with and without biocontrol activity. We revised the text in the methods and moved the original text to the results.

Point 7: Finally, why only five sequences of bacterial strains were analysed and not all nine of them?

Isolates that suppressed growth in vitro and lesion size in vivo were selected for sequencing

Point 8: Results are presented in a clear and convincing manner. Concerning phylogenetic analysis, the sequences of tested strains should be provided (possibly in Supplementary materials) or at least deposited in GenBank database to show Accession Numbers on the dendrogram. Otherwise, it is impossible to verify this analysis.

Sequences were deposited in GenBank. Accession numbers have been added to the text.

Round 2

Reviewer 3 Report

The revised manuscript was significantly improved. All of reviewer's suggestions were addressed properly.